# A Dual-Adaptive Approach Based on Discrete Cosine Transform for Removal of ECG Baseline Wander

Chun-Chieh Lin [1,*], Pei-Chann Chang [2] and Ping-Heng Tsai [3]

1   Department of Information Management, National Taipei University of Business, Taipei 100025, Taiwan
2   School of Accounting and Finance Beijing Institute of Technology, Zhuhai 519085, China
3   Department of Finance, Chihlee University of Technology, New Taipei City 220, Taiwan
*   Correspondence: leonjye@ntub.edu.tw

**Abstract:** Removal of baseline wander (BW) is an important preprocessing step before manually or automatically interpreting electrocardiogram (ECG) records. It is a challenging issue to fully remove BW while preserving original clinical information because BW is usually mingled with low-frequency ECG components. A dual-adaptive approach based on discrete cosine transform (DCT) is presented in this study. Firstly, the cardiac fundamental frequency (CFF) of ECGs is accurately calculated through DCT domain analysis. Secondly, DCT coefficients of ECGs, whose frequencies are below CFF, are used to construct an amplitude vector in which the optimal cut-point between BW and ECGs is distinctly reflected. Finally, a new filtering technique based on DCT is exploited to suppress BW with its cutoff frequency adjusted to the optimal cut-point. The proposed method is applied to both real ECG records and simulated ECGs with its results compared to those of three previous methods published in the literature. The experimental results show that substantial improvements in performance can be achieved when adopting this dual-adaptive approach.

**Keywords:** ECG preprocessing; baseline wander; discrete cosine transform

## 1. Introduction

With the development of computing technology, computer-aided diagnosis of cardiovascular diseases based on electrocardiograms (ECGs) has emerged as a popular area of research [1–3]. However, the ECG signal is prone to being contaminated with artifacts because it is captured and externally recorded by skin electrodes. Baseline wander (BW) is a type of low-frequency artifact in ECG which may stem from respiration and body movements. Removal of BW is an important step in the preprocessing stage of the ECG signal.

However, the spectrum of BW usually overlaps with those of low-frequency ECG components such as the T-wave, P-wave, and ST segment. Removal of BW can cause distortions to these components which contain important clinical information. For example, the ST segment is a typical measurement for the diagnosis of ST-elevation myocardial infarction (STEMI) [4,5], while microvolt T-wave alternans (MTWA) has a strong association with the increased risk of ventricular tachyarrhythmias [6]. Therefore, it is desirable that any algorithm used for BW removal can fully eliminate BW and also preserve the original clinical information.

In the past three decades, many approaches have been reported in the literature to address this issue. In general, these methods can be categorized as follows: linear filtering methods, nonlinear filtering methods, curve-fitting methods, and transform-reconstruction methods.

Linear filters, whose output is a linear function of the input, mainly contain two types: finite impulse response (FIR) filters and infinite impulse response (IIR) filters. FIR filters are normally preferred because of their linear phase characteristics [7], but a major drawback of these methods is that their cutoff frequencies are time-invariant. Nonlinear filters such as weighted median filters (WMFs) [8] and morphological filters [9,10] are receiving increasing

attention when ECGs and BW spectra overlap. However, the WMF-based method has a high computation cost because samples within its sliding window need to be maintained in sorted order all the time. Whereas, the methods based on morphological filters must design a proper structuring element for morphological operators.

Curve fitting is another nonlinear approach for BW correction. Cubic splines, polynomial, and rational functions are used to fit the BW after PR-segments have been determined [11,12] while [13] suggest that when the magnitude of BW is small, it is better to approximate by linear functions. However, the two biggest hurdles associated with curve-fitting methods are the extraneous frequency components brought into the ECG signal and the characteristic-segment detection requirement.

Transform-reconstruction methods, as their name suggests, contain two steps. Firstly, the transformation of the raw ECG signal into a transform domain in which the coefficients corresponding to BW are eliminated. Secondly, the reconstruction of the output signal, which is BW corrected, from the residual coefficients. Methods based on principal component analysis (PCA), independent component analysis (ICA) [14], wavelet transform (WT) [15–17], and discrete cosine transform (DCT) [18] fall into this category. WT-based methods assume that BW can be captured by the coarse approximation $cA_j$ of level $j$. Therefore, how to choose the proper level, $j$, emerges as a crucial problem [19]. DCT is used for the estimation of BW by making the higher-order DCT coefficients zeroes [20]. This method has proven to be effective since it can be regarded as a linear-phase low-pass filter. However, the cut-point between BW and the inherent ECG components is determined simply by detecting the peak in the Fourier spectrum.

In this paper, an improved DCT-based method is proposed, which adopts a dual-adaptive scheme to search for the optimal cut-point between BW and inherent ECG components before conducting BW removal. The remainder of the paper is organized as follows. Section 2 briefly introduces the theoretical background of DCT and then proposes the dual-adaptive scheme. Experimental results are illustrated in Section 3. Finally, Section 4 concludes the article.

## 2. Methods

### 2.1. Theoretical Backgrounds of Discrete Cosine Transform

DCT was first defined and implemented by using fast Fourier transform (FFT) in 1974 [21]. It was often used in feature extraction [22] to generate better classification. Equations (1) and (2) reveal DCT and inverse discrete cosine transform (IDCT), respectively.

$$y(k) = w(k) \sum_{n=1}^{N} x(n) \cos(\pi(2n-1)(k-1)/2N), \ k = 1, 2, \dots, N \tag{1}$$

$$x(n) = \sum_{k=1}^{N} w(k)y(k) \cos(\pi(2n-1)(k-1)/2N), \ n = 1, 2, \dots, N \tag{2}$$

$$where \ w(k) = \begin{cases} 1/\sqrt{N} \ if \ k = 1 \\ \sqrt{2/N} \ if \ 2 \le k \le n \end{cases} \tag{3}$$

$N$ is the length of both time-domain data $x$ and DCT-domain data $y$. Given the sampling frequency $F_s$, the frequency difference between two adjacent DCT coefficients is $(F_s/2N)$ Hz. Equation (4) describes the correlation between the frequency and the DCT-index.

$$k = (N \times f)/(F_s/2) \tag{4}$$

where $f$ is the corresponding frequency of DCT-index $k$.

The properties of DCT are exploited to suppress electromyogram interference on ECGs [23]. A DCT algorithm can also be useful to extract the morphological characteristics from ECG signals and generate the coefficients in the preprocessing stage [24].

## 2.2. The Proposed Dual-Adaptive Scheme

It has been repeatedly emphasized that the determination of the cut-point between BW and inherent ECG components is of great importance in order to fully eliminate BW and preserve the original clinical information. To illustrate this, a 20 s ECG (Figure 1a) from the PTB Diagnostic Database [25] was considered. There were 30 heartbeat occurrences in the 20 s ECG signal. We then calculated the cardiac fundamental frequency (CFF), which is the counterpart of heart rate and indicates the frequency of heartbeat occurrence in units of time. The accurate calculation of CFF is important because it is recommended to have a cutoff frequency that does not exceed CFF when filtering BW in the ECG signal [26]. In Figure 1a, CFF is 1.5 Hz approximately. Unfortunately, no visible peak exists (Figure 1b) around 1.5 Hz in the Fourier spectrum because it is obscured by severe BW.

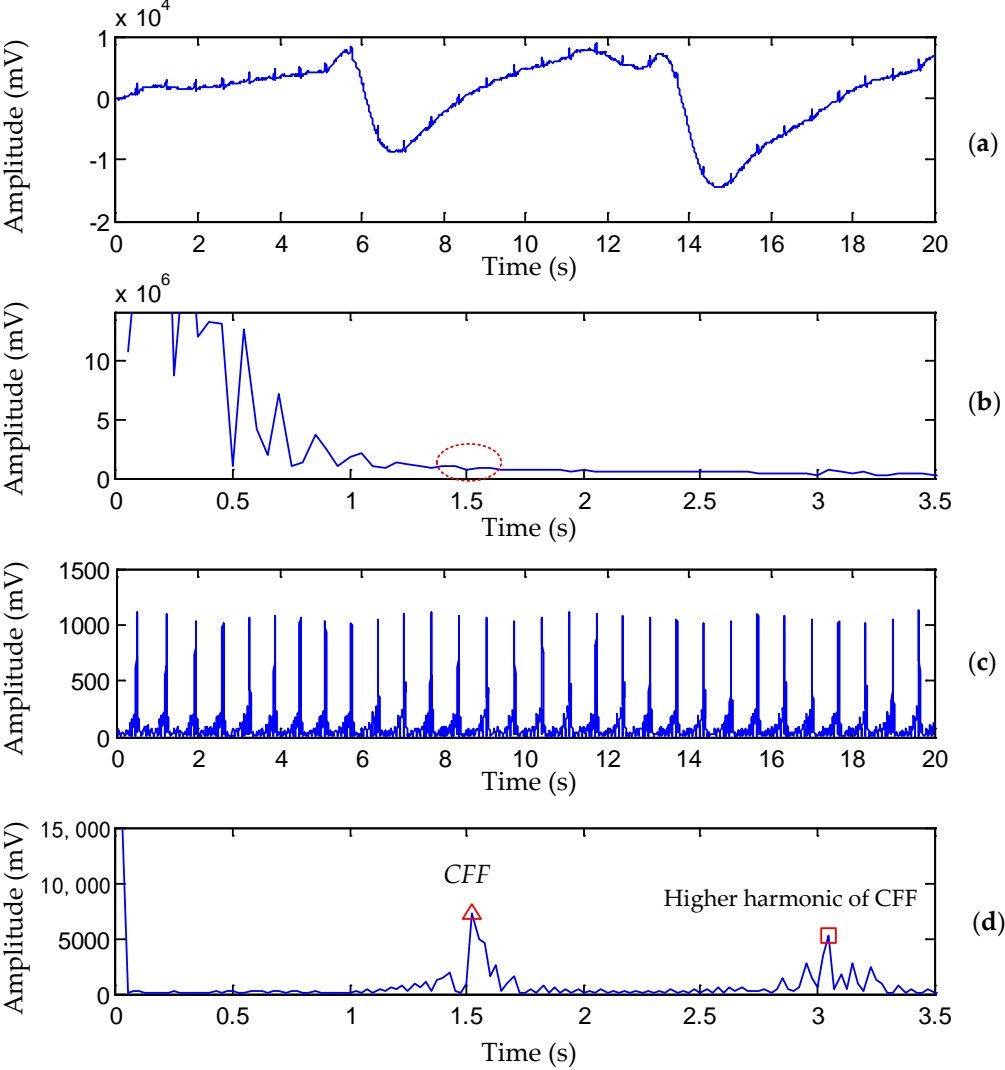

**Figure 1.** Comparison of CFF detection by the previous method [20] and the proposed method (PTB record s0130lrem, lead I, 00:00−00:20). (**a**) Raw ECG signal contaminated with severe BW. (**b**) Amplitude of raw ECG in the Fourier domain. (**c**) Waveform of the extracted QRS complexes $|x_{QRS}(n)|$. (**d**) Amplitude of $|x_{QRS}(n)|$ in the DCT domain, the red 'triangle' and red 'square' denote the position of CFF and its higher harmonic, respectively.

To overcome this drawback, we developed a novel scheme, called a dual-adaptive scheme (DAS), which can remove BW adaptively with the optimal cut-point not only adjusted to CFF, but also adjusted to the degree of BW existence. The proposed DAS contains three major steps as shown in Figure 2.

**Step 1.** Calculating the cardiac fundamental frequency (CFF)

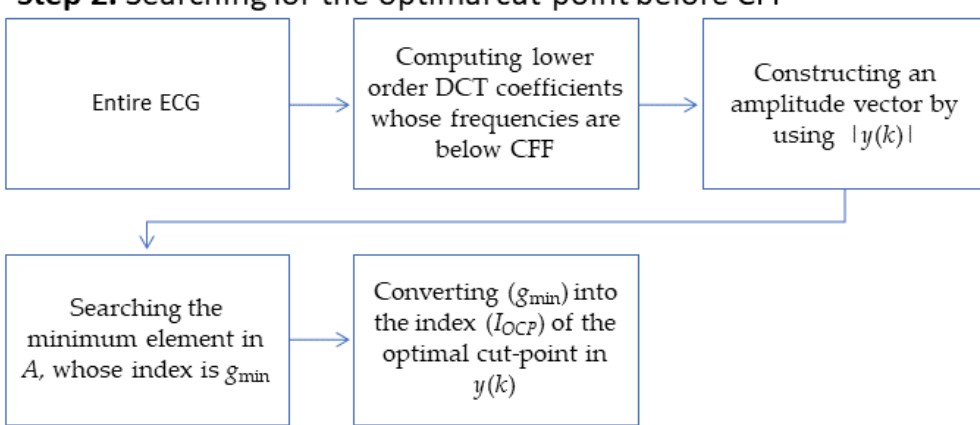

**Step 2.** Searching for the optimal cut-point before CFF

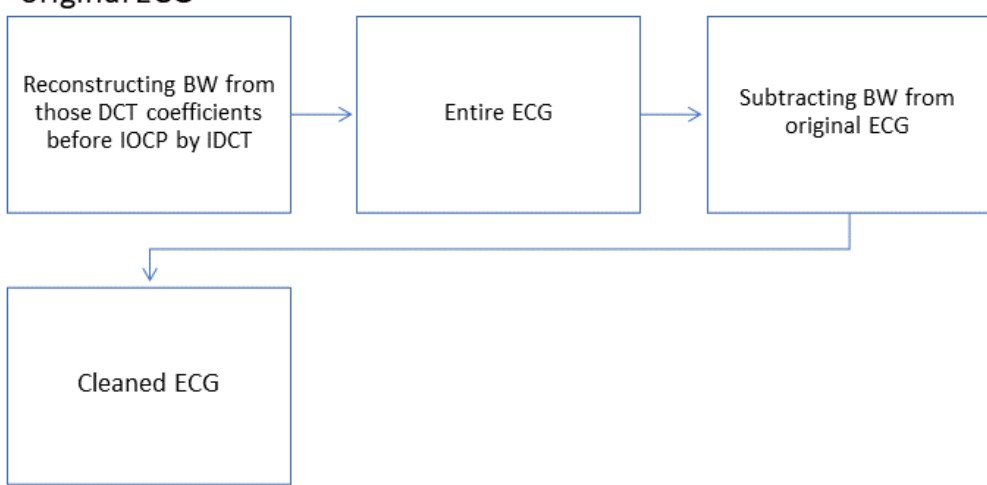

**Step 3.** Reconstructing baseline wander and subtracting it from original ECG

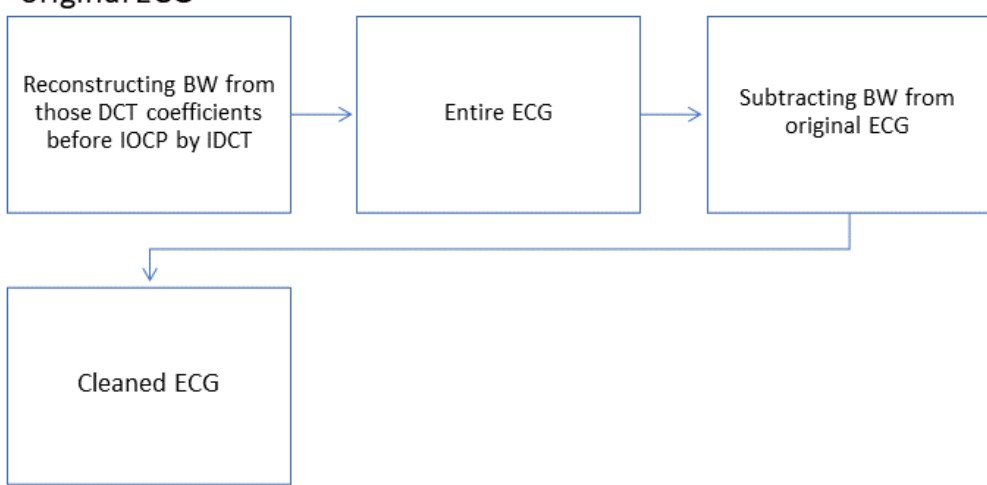

**Figure 2.** Block diagram of the Dual-Adaptive Scheme.

In the first step, CFF is calculated using a 20 s long portion of the ECG that is to be processed. In the second step, the optimal cut-point before CFF is investigated by analyzing the lower order DCT-coefficients. Finally, BW is reconstructed by applying IDCT to those DCT coefficients before the optimal cut-point, and the BW-free ECG is retrieved by subtracting BW from the original ECG. The detailed procedure is described in the sections below.

### 2.2.1. Calculating the Cardiac Fundamental Frequency (CFF)

Common methods to calculate CFF are based on R-wave detection. However, the R-wave detection itself is an awful task in the presence of various artifacts. Moreover, the CFF calculation according to the R-wave occurrence is improper in the case of abnormal ECG records, which is illustrated in Section 3. In this paper, we present a simple and effective method to calculate CFF through DCT domain analysis. It is well known that, for periodic waves, there must be a distinct peak that corresponds to the fundamental frequency in the spectrum domain. ECG signals can roughly be regarded as periodic waves, but their periodic properties are usually destroyed by artifacts. Nevertheless, the periodicity of ECG is well reflected by its QRS-complexes. Based on these ideas, CFF can be calculated by the following three substeps.

- Extract the QRS-complexes $x_{QRS}(n)$ from the ECG signal $x(n)$ by a DCT-based band-pass filter whose passband is [5~40] Hz.
- Waveform $|x_{QRS}(n)|$ in Figure 1c is taken from QRS-complexes to strengthen the periodicity and the DCT coefficients $|y_{QRS}(n)|$ (in Figure 1d) of $|x_{QRS}(n)|$ are calculated afterward.
- Detect the peak's index $I_{CFF}$ related to CFF in $|y_{QRS}(n)|$. Detecting $I_{CFF}$ is based on a threshold decision as shown in Figure 1d. The maximum value $V_m$ of $|y_{QRS}(n)|$ in the range of 0.2 Hz to 2.5 Hz is detected at first. The index $I_m$ of $V_m$ may be related to CFF but also could be related to the higher harmonics of CFF. In this study, a threshold value of 0.65 was chosen empirically to determine $I_{CFF}$ which is the first point whose value exceeds $(\theta * V_m)$.

### 2.2.2. Searching for the Optimal Cut-Point before CFF

According to Equation (2), ECG signals can be decomposed into a combination of cosine waves. An example of the decomposition of the low-frequency part of the ECG, whose frequency is below CFF, is shown in Figure 3. In Figure 3b, each cosine waveform is generated from one DCT coefficient, as shown in the following formula.

$$x_k(n) = w(k)y(k)\cos(\pi(2n-1)(k-1)/2N) \tag{5}$$

where $x_k(n)$ is the cosine waveform generated from the $k^{th}$ DCT coefficient $y(k)$.

These cosine waves could have overlapped with each other, but certain amounts of offset were added to separate them exactly for a good presentation. Two phenomena are illustrated in Figure 3b. Firstly, the cosine waves tended to have higher frequencies while the indexes of the DCT coefficient were increasing. Secondly, the waves near both the BW area and the CFF area had higher magnitudes, so the optimal cut-point to distinguish BW and the ECG could be regarded as a certain value in the area where the cosine waves bunched up together. The summation of those cosine waves below the cut-point yielded a trend signal (the red curve in Figure 3a), which was the ideal estimation of BW.

The goal of determining the optimal cut-point was transformed to search for the area where cosine waves were the flattest as shown in Figure 3b. Assume that there were a total number of $I_{CFF}$ waves whose frequencies were below CFF, where $I_{CFF}$ was calculated according to CFF by Equation (4).

These waves were uniformly divided into $G$ groups, therefore each group contained the same number $M$ of waves, where $M$ is the quotient of $I_{CFF}$ divided by $G$.

$$g_{min} = \arg min_g \sum_{k=(g-1)*M+1}^{g*M} 2 * A_k \; 1 \le g \le G \tag{6}$$

where $A_k$ is the amplitude of the $k^{th}$ cosine waveform, and $g$ denotes the group's index. As illustrated in Figure 3c, the way to determine where the optimal cut-point is located is to calculate the $g_{min}$ group which has the minimum value of amplitudes by Equation (6).

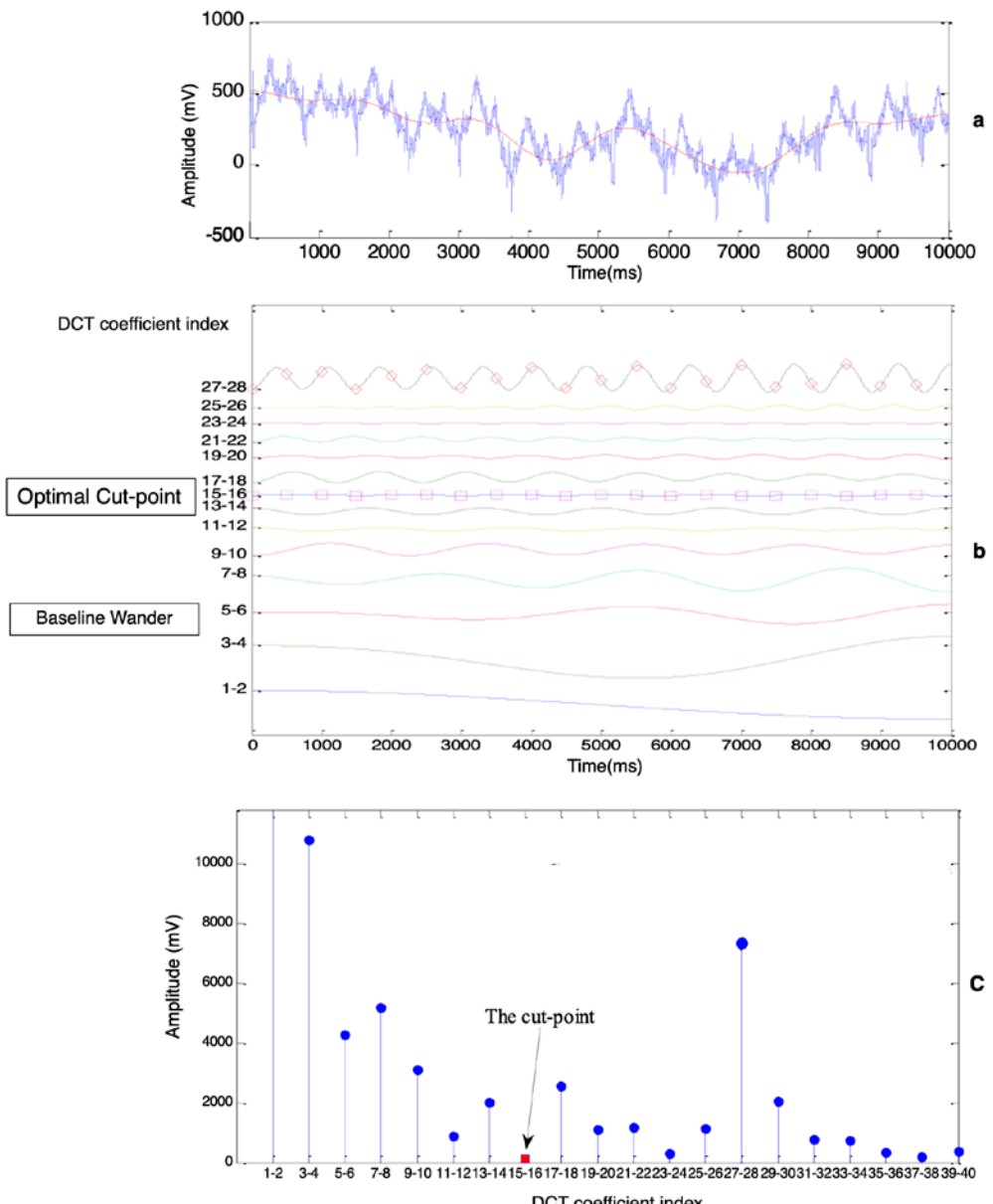

**Figure 3.** Determine the optimal cut-point between BW and ECG (lead AVF of PTB record patient004/s0020arem). (**a**) Raw ECG signal (blue curve) and BW (red curve). (**b**) Cosine waves, each of which is generated from one DCT coefficient. (**c**) Amplitude of low-frequency components, each component is generated from two DCT coefficients.

With Equations (3) and (5), we can deduce that:

$$A_k = |w(k)| * |y(k)| = \begin{cases} |y(k)|/\sqrt{N} & k = 1 \\ |y(k)| * \sqrt{2}/\sqrt{N} & 2 \leq k \leq N \end{cases} \tag{7}$$

Therefore, we obtain the equivalent of (6) without considering the constant:

$$g_{min} = \arg min_g \sum_{k=(g-1)*M+1}^{g*M} |y(k)| \; 1 \leq g \leq G \tag{8}$$

where $|y(k)|$ is the absolute value of the $k^{th}$ DCT coefficient.

Based on the above analysis, we searched the optimal cut-point directly according to Equation (8) by analyzing DCT coefficients whose frequencies were below CFF:

The lower order DCT coefficients of the ECG signal $x(n)$ were calculated according to the following equation.

$$y(k) = w(k) \sum_{n=1}^{N} x(n) \cos(\pi(2n-1)(k-1)/2N), \ k = 1, 2, \ldots, I_{CFF} \tag{9}$$

These coefficients were used to construct an amplitude vector $A = [a_1, a_2, a_3, \ldots, a_G]$, each element of $A$ was a summation of the same number $M$ of $|y(k)|$. The minimum element in vector $A$ was investigated to obtain its index, $g_{min}$. The index $I_{OCP}$ of the optimal cut-point in $y(k)$ was calculated from the following equation.

$$I_{OCP} = M * g_{min} \tag{10}$$

2.2.3. Reconstructing BW and Subtracting It from the Original ECG

The following equation was used to reconstruct BW from those DCT coefficients before the optimal cut-point.

$$bw(n) = \sum_{k=1}^{I_{OCP}} w(k)y(k) \cos(\pi(2n-1)(k-1)/2N, \ n = 1, 2, \ldots, N \tag{11}$$

then, a BW-free ECG could be restored by subtracting the estimated BW from the original ECG. It is shown as follows.

$$x_r(n) = x(n) - bw(n) \tag{12}$$

where $bw(n)$ is the estimated BW, $x_r(n)$ is the 'retrieved' ECG after BW removal, and $x(n)$ is the original ECG contaminated with BW.

**3. Results**

Experiments were carried out on both real ECG records and simulated ECGs to evaluate the proposed method qualitatively and quantitatively. The real ECG records were from the MIT-BIH Arrhythmia Database and the PTB Diagnostic Database. The former database is a collection of 48 half-hour two-lead ECGs whose sampling rate is 360 Hz. The latter is a database of 549 high-resolution 15-lead ECGs digitized at 1000 samples per second. The simulated "clean" ECG was constructed by repetitively concatenating an identical ECG cycle chosen from real ECG records. The simulated BW was generated by calculating low-frequency sinusoid waves and a low-frequency random component [13].

For comparison, the performances of the following five algorithms were investigated:

1. A DCT-based filter [27] with the proposed dual-adaptive scheme to search for the optimal cut-point between BW and the true ECG. ($\theta = 0.65$, $G = 10$).
2. A DCT-based filter [27] which let the cutoff frequency be $0.9 \times$ CFF.
3. A linear phase, sharp cut-off FIR filter [28] with a cutoff frequency of $0.9 \times$ CFF.
4. A WT-based algorithm using Daub-4 mother wavelet [29].
5. A weighted median filter [30] with parameters ($a = 500$, $\alpha = 1$, $\beta = 40$).

The algorithms are referred to using Roman numerals I–V.

*3.1. Performance Comparison on Real ECG Record*

In the first experiment, to illustrate the effectiveness of the proposed method on the CFF calculation, which is a vital step for the whole process, the first 20 s portion of each record in the MIT-BIH database was taken into consideration. For the sake of comparison, the "gold standard" CFF should have been calculated according to its definition, "CFF is the frequency of heartbeat occurrence in unit of time". However, the number of heartbeat occurrences is difficult to count in a 20 s portion due to the incompleteness of the heartbeat at each end. The "gold standard" CFF is approximately calculated using Equation (13) according to R-wave occurrence, which is an analog of heartbeat occurrence and much easier to count.

$$CFF_{R-wave} = N_{R-wave}/L \tag{13}$$

where $N_{R-wave}$ is the number of R-waves that occur in $L$ seconds-long ECG, and $CFF_{R-wave}$ is the approximation of the "gold standard" CFF according to R-wave occurrence. The correctness of the CFF calculation with the proposed method was measured using the following Equation (14).

$$|CFF_{dct} - CFF_{R-wave}| < 0.1 * CFF_{R-wave} \tag{14}$$

where $CFF_{dct}$ is the CFF calculated using the proposed method through DCT domain analysis. $CFF_{dct}$ was considered to be correct if it tallied with this rule. It was reasonable to adopt this rule because the $CFF_{R-wave}$ itself may have a maximum difference of $(0.1 * CFF_{R-wave})$ with the real "gold standard" CFF.

Experimental results demonstrated that, only 4 (record number: 203, 208, 221, and 232) out of a total of 48 records in this database failed to meet the rule, which means that the proposed CFF-calculation method achieved an accuracy of 91.67%. The four failed records were abnormal cases that contained ectopic heartbeats as shown in Figure 4. As a supplement, the same experiment was conducted with the PTB database and it obtained a higher accuracy of 95.80%.

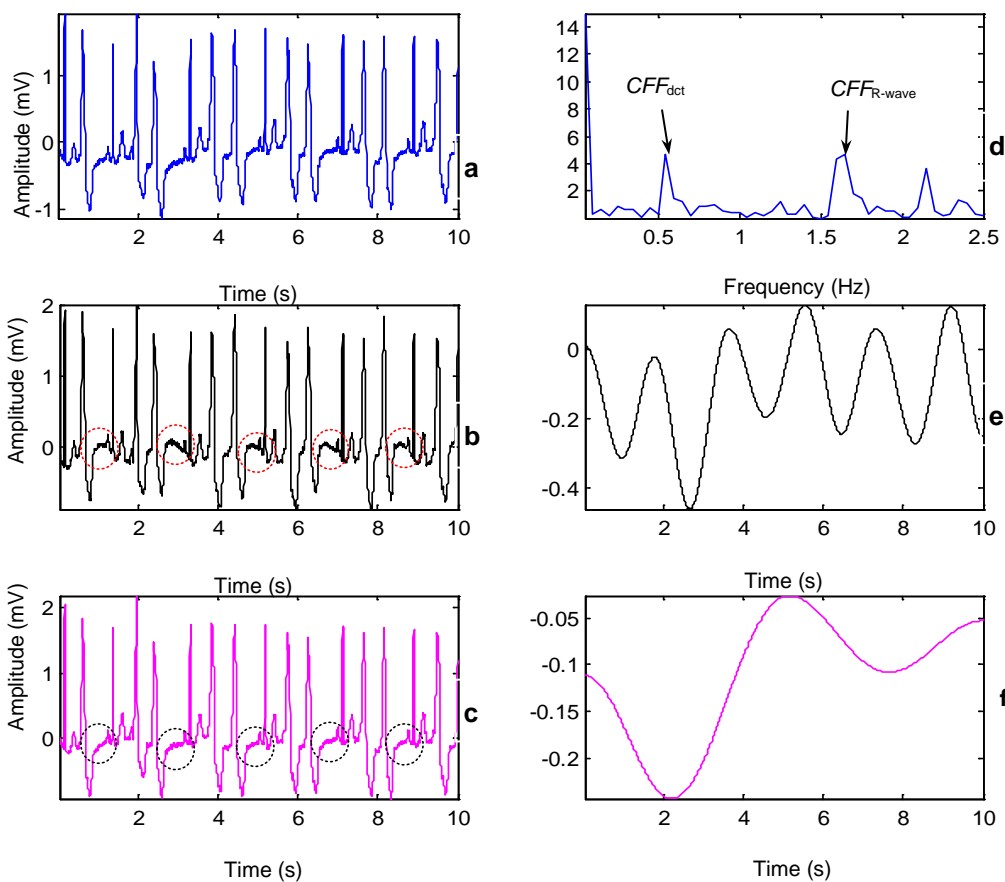

**Figure 4.** The impact of two CFFs when processing abnormal records (MIT-BIH record 208, lead I, 00:00–00:10). (**a**) Original ECG. (**b**,**c**) BW corrected ECG by using $CFF_{R-wave}$ and $CFF_{dct}$, respectively. (**d**) Locations of the $CFF_{R-wave}$ and $CFF_{dct}$ in the DCT domain. (**e**,**f**) Estimated BWs by using $CFF_{R-wave}$ and $CFF_{dct}$, respectively. The surrounding ellipses indicate the ST segment of the ectopic beats.

In the second experiment, BW removal was carried out on abnormal records using $CFF_{R-wave}$ and $CFF_{dct}$, respectively, in order to compare the impact of the two CFF-calculation schemes on the final result. After the CFF ($CFF_{R-wave}$ or $CFF_{dct}$) had been calculated, a DCT-based filter with a 0.9×CFF cutoff frequency was applied.

The comparison results of the two CFFs is demonstrated in Figure 4. BW removal using the two CFFs was conducted on the abnormal record 208 which included premature ventricular contractions (PVCs). It was observed that the reconstructed ECG using $CFF_{R-wave}$ had some distortions especially on the ST segment (which is framed in ellipses in Figure 4b). The ST-segment elevation was one of the abnormal presentations in patients with PVCs [31,32], if we compared the ECGs with the first few heart heats in detail shown in Figure 5, the original ECG in Figure 5a presented the ST-segment elevation, while the ECG in Figure 5b may have suppressed the important information. In contrast, the proposed method based on DCT-domain analysis achieves a promising result with keeping the ST-segment elevations shown in Figure 5c. The reason behind this is that the periodicity of ECG signals may be destroyed by abnormalities such as PVCs and arrhythmias. As Figures 4b and 5b show, there was a prominent unusual frequency component, caused by PVCs that appeared before $CFF_{R-wave}$. Note that it still belongs to inherent ECG components and eliminating it would consequently cause distortions. However, it cannot be discovered using the CFF-calculation method based on R-wave occurrence and thence is prone to be eliminated. Fortunately, the proposed method is able to detect this unusual component and 'mistake' it for CFF so as to keep it from being removed.

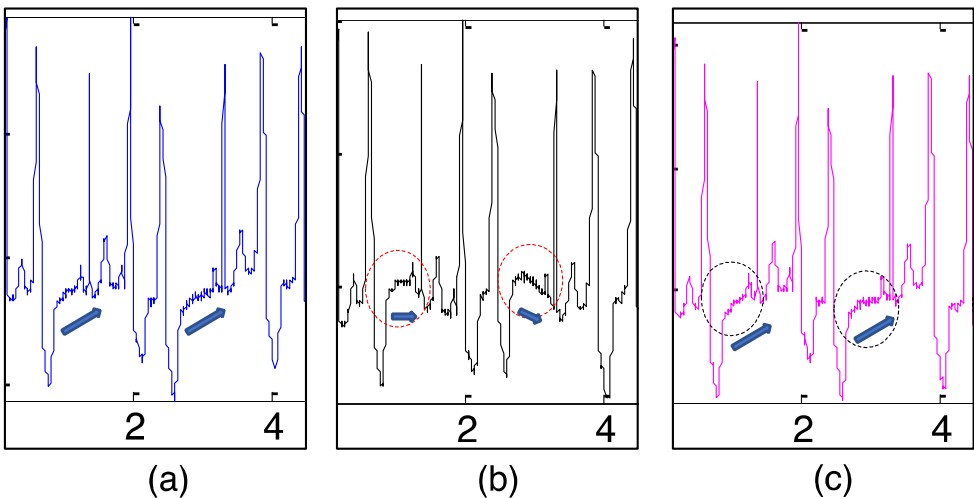

(a)    (b)    (c)

**Figure 5.** The comparison in detail from Figure 4, with the first few heartbeats included. (**a**) Original ECG in Figure 4a. (**b**) BW corrected ECG using the $CFF_{R-wave}$ in Figure 4b. (**c**) BW corrected ECG using the $CFF_{dct}$ in Figure 4c.

In the third experiment, a comparison between the proposed method and the WT-based filter was made. Consecutive numbers eight, nine and ten were respectively chosen as the decomposition level of the WT-based method. The reconstructed BWs using different decomposition levels were plotted overlapping with the original ECG in Figure 6a–c. As Figure 6a shows, when the decomposition level was eight, the reconstructed BW was overfitting which indicated that the decomposition level needed to be higher. However, when the decomposition level was 10, which is too high, as shown in Figure 6c, the reconstructed BW was insufficient to fit the tendency of the ECG, especially in the surrounding areas highlighted by cycles. We obtained the optimal decomposition level nine between eight and ten and the best estimated BW is shown in Figure 6b. Then, the retrieved ECG using the WT-based method at the optimal decomposition level was compared to the one using the proposed method. As Figure 6e shows, although the optimal decomposition level was chosen, the retrieved ECG using the WT-based method still had some distortions in those areas surrounded by ellipses. In contrast, as shown in Figure 6f, it is apparent that BW was fully removed using the proposed method without introducing visible distortions.

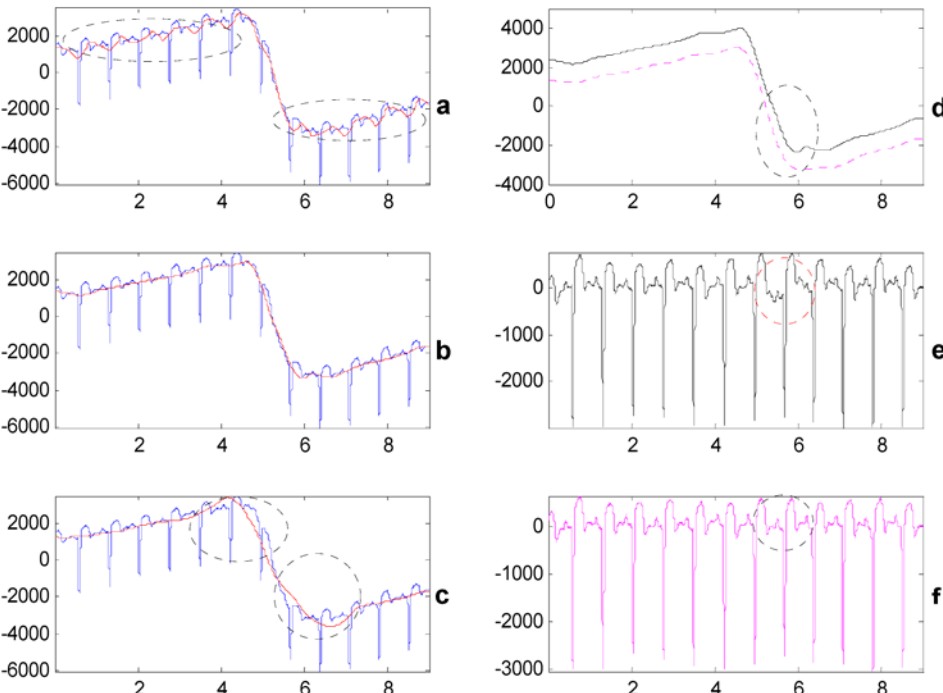

**Figure 6.** Comparison between the proposed method and the WT-based method (PTB record s0098lrem, lead V2, 00:10–00:19). (**a**–**c**) The blue waveform is BW contaminated ECG, while the red curve is the estimated BW using algorithm IV when wavelet decomposition levels are 8, 9, and 10. (**d**) The black solid curve is the estimated BW using algorithm IV when wavelet decomposition level is nine (1000 µV offset is added), and the magenta dotted curve is the estimated BW using algorithm I. (**e**,**f**) BW corrected ECG using algorithm IV (decomposition level = 9) and algorithm I, respectively.

In the fourth experiment, an ECG signal contaminated with severe BW was deliberately chosen in order to validate the proposed method when handling severe BW. Figure 7 shows the original ECG and the retrieved ECGs using five algorithms in a list fashion. By scrutinizing the middle part of each retrieved ECG, it is clear that BW was not fully removed using algorithms III and V. As for algorithms II and IV, although they fully removed BW, they also introduced visible distortions. The proposed method (algorithm I) achieved the best performance that fully removed BW while preserving the original clinical information.

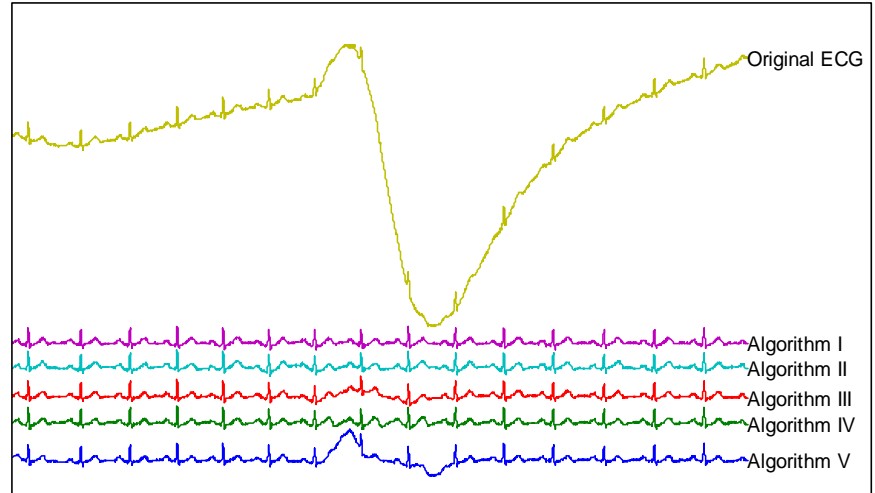

**Figure 7.** Comparisons of the five algorithms for suppressing severe BW (PTB record, s0130lrem, lead AVF, 00:10–00:21).

### 3.2. Experiments with Simulated ECG and BW

In order to evaluate the performances of different methods quantitatively, tests should be conducted on simulated ECG and BW whose properties are already known. We constructed the 'clean' ECGs by concatenating identical beats chosen from the PTB database. In total, 24 clean ECG signals were simulated. The length of each signal was 30 s, and the sampling rate was 1000 Hz. The beats of these signals had different morphological features and heart rates. They were chosen from 12 different leads of different records. The simulated BW, given in Equation (15), consisted of four components. The sinusoid components were used to mimic the typical BW caused by respiration [13]. Low-frequency random noise was also added in order to simulate the random noise from the outside environment. This random component was generated by filtering white noise (700 μV amplitude) through a fifth-order Butterworth low-pass filter with 0.3 Hz cutoff frequency. With the last random components generated 30 times, 30 different BWs were obtained.

$$bw(n) = A * \sin(2\pi * 0.2n/F_s) + B * \cos(2\pi * 0.35n/F_s) + C * \cos(2\pi * 0.45n/F_s) + \eta(n) \quad (15)$$

where $\eta(n)$ is the low-frequency random component and *A*, *B*, and *C* are amplitudes of sinusoid components.

In the last experiment, the five algorithms were applied to the simulated ECGs. One example of simulated clean ECG and BW is shown in Figure 8. The contaminated ECG was obtained by summing the simulated clean ECG and BW. The reconstructed BWs using five algorithms were also plotted in this figure. As Figure 8 shows, it was difficult to distinguish the BWs reconstructed using algorithms I and II from the simulated one, which indicates that the DCT-based filters were capable of estimating the added BW ideally. Compared with the simulated BW, the one reconstructed using algorithm III introduces significant unwanted oscillations which were higher frequency components that came from 'clean' ECGs. Algorithms IV and V achieved moderate performances but nonetheless introduced visible distortions. Additionally, BW constructed using algorithm V was not smooth although the simulated BW was a smooth one, which is due to the nonlinear characteristic of this filtering technique.

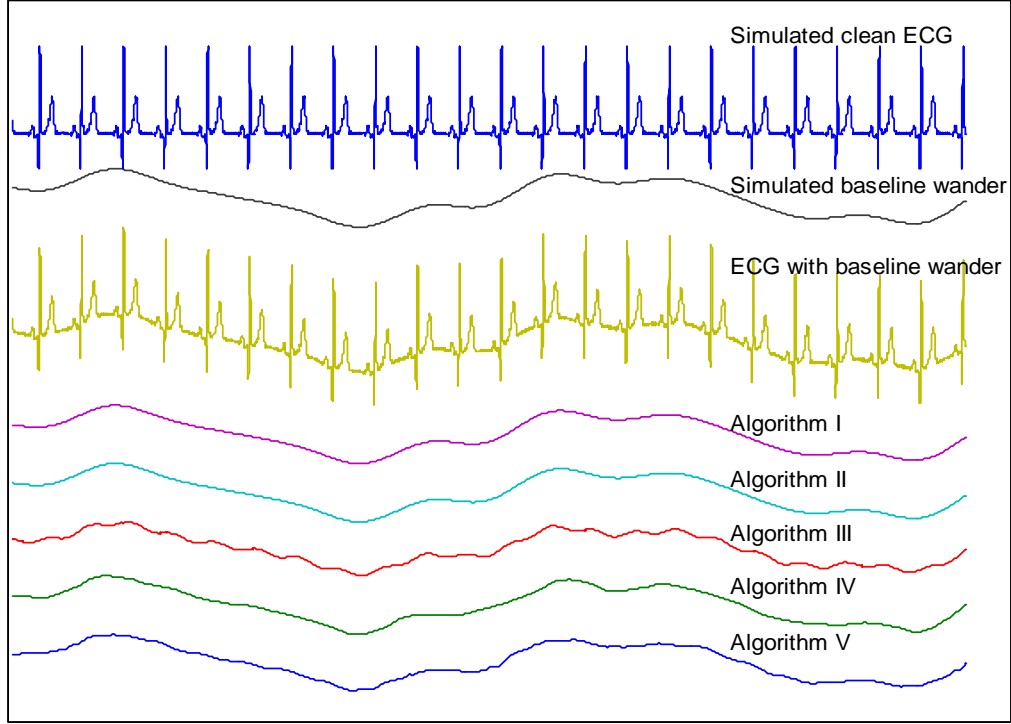

**Figure 8.** Comparison of estimated BW using five algorithms using the simulated ECG and BW.

For quantitative evolution, the most widely used measures are mean square error (MSE), and signal-to-noise ratio (SNR). However, using these measures, it is difficult to access the quality of filtered signals which have different mean values due to the DC offset. All algorithms in this paper generated zero-mean 'retrieved' ECG signals except algorithm V. In order to compare these algorithms fairly, we calculated the correlation coefficient (CC) between simulated 'clean' ECGs and 'retrieved' ECGs to evaluate their performances. It can be seen from Equation (16) that CC was independent of the signals' mean values because there was an inner process to subtract the mean value of each signal from itself. The higher CC indicates the better properties of algorithms in preserving the original clinical information.

$$\rho_{cr} = \sum_{m=1}^{N} [x_c(n) - u_c][x_r(n) - u_r]/\sigma_c \sigma_r \tag{16}$$

where $x_c(n)$ is the 'clean' ECG by simulation, $x_r(n)$ is 'retrieved' ECG by BW removal algorithms, $\rho_{cr}$ is CC between these two signals. $u_c$ and $u_r$ are the expected values of $x_c(n)$ and $x_r(n)$, respectively, and $\sigma_c$, $\sigma_r$ are the standard deviations of $x_c(n)$ and $x_r(n)$, respectively.

As mentioned before, a total of 24 'clean' ECGs and 30 BWs were generated. After applying five algorithms to these simulated ECGs, the CC of each 'retrieved' ECG with respect to the simulated 'clean' ECG was calculated. But the retrieved ECGs using algorithms III and V have significant distortions at each end of the signal due to the moving window used in these techniques. Therefore, for all retrieved signals, the first and last 500 samples were not considered when calculating CC in order to make a fair comparison. The comparison results are plotted in Figure 9, in which each point represents an average of 30 experiments. Two major aspects were revealed in Figure 9. First, it is striking that the performances of algorithms I and II were stable and far superior to those of the other three algorithms. Second, the CC curves of algorithms III, IV, and V fluctuated a lot from point to point, which indicates that their performances were sensitive to the morphological features and heart rate of the simulated ECG.

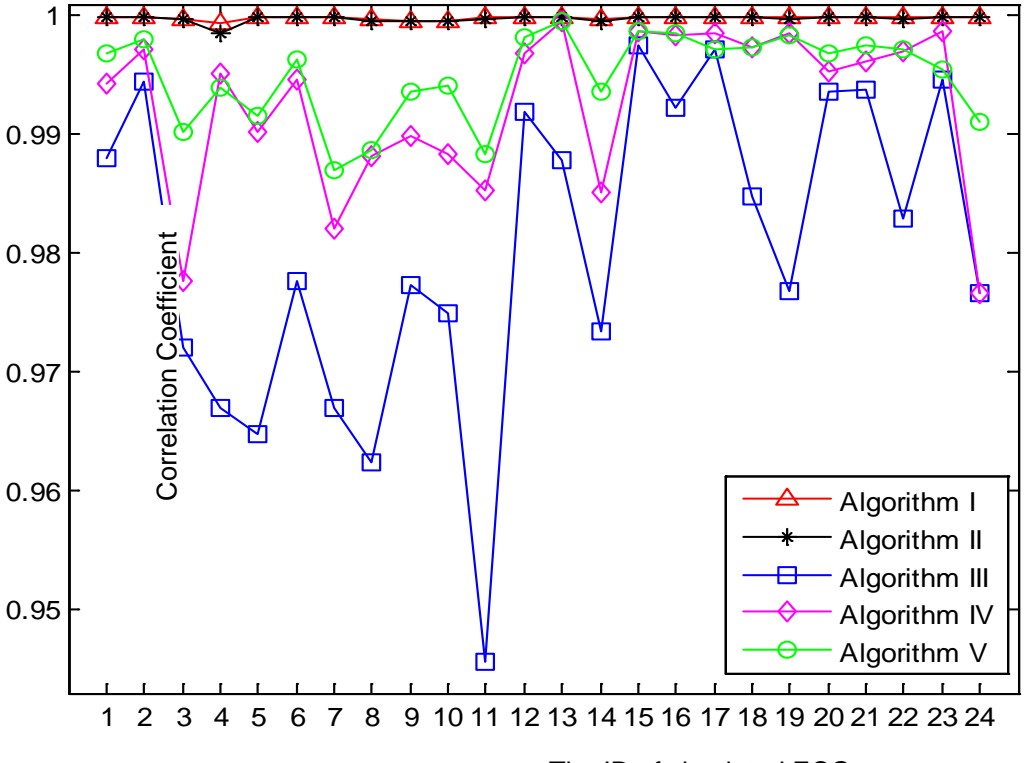

**Figure 9.** Comparison of five algorithms using the correlation coefficient between 'retrieved' and 'clean' ECG.

## 4. Discussion

Benefitting from the robust CFF-calculation subprocedure, both algorithms I and II achieved the most promising results. Nonetheless, algorithm I slightly outperformed algorithm II as shown in Figures 7 and 9. One major factor that contributed to the robustness of the proposed CFF-calculation method lies in the QRS-complexes extraction by the DCT-based band-pass filter. Compared with other CFF-calculation methods based on R-wave detection, the proposed method illustrated its greater simplicity and effectiveness in the presence of various artifacts and abnormalities as shown in Figures 4 and 5.

The DCT-based filtering technique [27] was also compared to the linear phase, sharp cut-off FIR filter [28]. Both types of filters had a linear-phase response which was their common favorable characteristic for BW removal. As shown in and Figure 8, some inherent ECG components are filtered by the FIR filter, while the DCT-based filter seldom affects the inherent components although their cutoff frequencies are set to be the same. This is because the DCT-based filter possesses unique ideal frequency characteristics. For instance, the transition band of the DCT-based filter can be as narrow as $(F_s/2N)$ Hz theoretically, which is the frequency interval between two adjacent DCT coefficients. Moreover, the DCT-based filter has negative infinite magnitude response on the stop band and infinite passband gain theoretically [27]. However, these characteristics cannot be realized by the FIR filter because it needs an infinite impulse response. Therefore, the DCT-based filter outperforms the FIR filter in terms of eliminating the unwanted frequency components precisely with the inherent ECG components intact.

As shown in Figures 6–9, the proposed method is superior to the WT-based method. This is because WT-based methods are based on the assumption that BW can be captured by the coarse approximation $cA_j$ of level $j$. The frequency span of $cA_j$ is $\left[0 \sim (F_s/2) * 2^{-j}\right]$, which indicates that the estimated BW has the cutoff frequency of $(F_s/2) * 2^{-j}$ Hz. For example in Figure 6, if $j$ is chosen to be 8, 9, and 10, then the cutoff frequency of the estimated BW is 1.9531, 0.9766, and 0.4883 Hz, respectively. Evidently, it is not possible to adjust the cutoff frequency precisely in accordance with CFF and the degree of BW exists in ECG. In contrast, the proposed method seeks for the optimal cut-point by constructing an amplitude vector whose frequency span is CFF. Given that the amplitude vector contains G elements, then the frequency resolution of the proposed method can be accurate to CFF/G. Therefore, the proposed method can remove BW more precisely than the WT-based method.

Finally, as shown in Figures 7–9, the proposed method outperformed the weighted median filter. The weighted median filter was unable to trace the turning corner caused by severe BW (shown in Figure 7) due to the improper length of its sliding window. However, the weighted median filter achieved better results than the WT-based method and FIR filter when processing ECGs with moderate BW in the simulated experiments. Nonetheless, the proposed method is far superior to the weighted median filter as illustrated by the CC curves in Figure 9.

In conclusion, the proposed algorithm achieved the best performance in terms of fully removing BW and preserving the original clinical information when compared with three previous methods. The reasons for its outstanding performance are threefold. Firstly, it contained a robust subprocedure to calculate CFF properly, which ensured that the cup-point between BW and the ECG did not exceed CFF. Secondly, it included a novel scheme to seek the optimal cut-point by means of constructing an amplitude vector using lower-order DCT coefficients. Thirdly, it adopted a new filtering technique based on DCT, which had the ideal frequency properties. Based on the above reasons, the unnecessary frequency components can be precisely eliminated while keeping the inherent ECG components intact.

**Author Contributions:** Conceptualization and methodology, C.-C.L.; validation, P.-H.T.; formal analysis and investigation, P.-C.C.; writing—original draft preparation, C.-C.L.; writing—review and editing, C.-C.L. and P.-H.T.; supervision and project administration, P.-C.C. All authors have read and agreed to the published version of the manuscript.

**Funding:** This research was funded by National Science and Technology Council of Taiwan grant number 111-2221-E-141-008-.

**Institutional Review Board Statement:** Not applicable.

**Informed Consent Statement:** Not applicable.

**Data Availability Statement:** The data presented in this study are openly available in PhysioNet at https://doi.org/10.13026/C28C71 and MIT-BIH Arrhythmia Database at https://doi.org/10.13026/c2f305.

**Conflicts of Interest:** The authors declare no conflict of interest.

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
