# Peer review of "A Dual-Adaptive Approach Based on Discrete Cosine Transform for Removal of ECG Baseline Wander"

_applsci, doi:10.3390/app12178839_

Round 1

Reviewer 1 Report

This paper proposes a dual adaptive filtering scheme using the DCT to remove baseline wander from recorded ECG signals. The competing algorithms compared are very old and it is not clear if the proposed algorithm hold any relevance in current times.

1. Page 2, line 81, please correct the missing equation numbers.

2. Fig. 1 (d), please correct the missing term in the caption. 'Amplitude of ____ in DCT'

3. Please improve the quality of Figure 2. Why are the red lines above the variables x_QRS and y_QRS?

4. The proposed algorithm is compared with algorithms II-V. These algorithms were proposed/reported in references 26-29, which are very old. Minimum of 12 years old and a maximum of 37 years old. Such comparison does not show the improvement the algorithm can achieve over state of the art algorithms. Hence, to justify the relevance of the proposed algorithm in today's date, the proposed algorithm should be compared with latest algorithms.

Author Response

Thanks for the reviewer's careful reading and the help to improve this paper.

  1. Page 2, line 81, please correct the missing equation numbers.
    Response 1: Thanks for the reminder, we already fix the errors and scrutinize all text in pdf files.
  2. Fig. 1 (d), please correct the missing term in the caption. 'Amplitude of ____ in DCT'
    Response 2: Thanks for the reminder again, we indeed missed the certain word in caption and added it back.

  3. Please improve the quality of Figure 2. Why are the red lines above the variables x_QRS and y_QRS?
    Response 3: Thanks for pointing it out. We redrew Figure 2 again. The reason of why the variables have red line above is the spell check in Word and we just take a screenshot carelessly.

  4. The proposed algorithm is compared with algorithms II-V. These algorithms were proposed/reported in references 26-29, which are very old. Minimum of 12 years old and a maximum of 37 years old. Such comparison does not show the improvement the algorithm can achieve over state of the art algorithms. Hence, to justify the relevance of the proposed algorithm in today's date, the proposed algorithm should be compared with latest algorithms.
    Response 4: Thanks for the suggestion. We gathered the newer papers which have the same or similar approaches and replaced some older papers with them. The number of newly added references are 27, 29 and 30.

Reviewer 2 Report

This paper addresses the problem of removing baseline wander from ecg signals.  The basic idea seems reasonable and if the authors can explain their method I would recommend publication.  However, they really need to do some work to explain what they are doing.

Their algorithm has three steps: Calculating the cardiac fundamental frequency, searching for the optimal cutoff for their DCT based filter below this fundamental frequency, and reconstructing the baseline wander and subtracting it from the original ecg.

The first step is mostly understandable though the description still needs work.  From what I understood they keep the DCT coefficients corresponding to the range between 5-40Hz.  They then reconstruct the signal and take the absolute value of the signal (which they keep referring to as modulus - please do not do this).  They then take the DCT of this signal and identify the peak in the absolute values of the coefficients corresponding to signals between 0.2 - 2.5 Hz. 

I think this is what they are doing but I am not sure because the writeup is not very clear.

In the second step I have no idea what they are doing.  The talk about the "area where the cosine waves are most dense." What do they mean by this?  They refer to a figure where the cosine waves are plotted with an unspecified offset for each.  Why on earth are you plotting the cosine waves?  Why not just plot the coefficients if you are interested in demonstrating the relative magnitudes of the coefficients?  And what do you mean by "dense?"  Please clean up this portion.  The idea of a paper is that you are telling others what you have done.  These are not notes to yourself.  I for one did not understand what you are doing here.

In the results section you need to actually read what you have written.  Just as an example consider the figure caption for Figure 4: "(c) BW corrected ECG using CFF_{dct} and CFF_{dct} respectively"  I am assuming one of these uses the fundamental frequency obtained using your method and other uses the fundamental frequency using the R-wave method.  Which is which is not clear.  This lack of proof-reading is really harmful to your description. 

You seem to feel that the "notable distortions, especially on ST segments" is clear from the figures.  It is not.  If you want to show distortions show a magnified view of the ST region.  You have the same problem with Figure 5.

Some minor issues:

Line 81 - the equation numbers show up as Chinese text.

Please define CFF before you use it.

Round 2

Reviewer 1 Report

I would suggest the authors to make the block diagram/signal flow in powerpoint and export as an image (preferably .tiff format) and add it to the manuscript. Also, the now the figure is repeated twice once in page 4 and once in page 5. Please correct this and modify the figure. Hence I am suggesting minor revision.

Author Response

Point 1: I would suggest the authors to make the block diagram/signal flow in powerpoint and export as an image (preferably .tiff format) and add it to the manuscript. Also, the now the figure is repeated twice once in page 4 and once in page 5. Please correct this and modify the figure. Hence I am suggesting minor revision.

Response 1: Thanks for the suggestion, we redrew figure 2 again by powerpoint with smartArt tool, and we did not find figure 2 was repeated in page four and page five, neither in word/pdf file format. We will check it again and submit the revised version.

Reviewer 2 Report

 The authors seem to see this process as adversarial; it is not. The goal of the review is to improve the paper.

1. The use of the word "modulus" is misleading in this context.  Use the phrase "absolute value" in Figure 2.  It will make your method more understandable.

2. In Figure 3 you say that "These cosine waves could have been overlapped with each other, but certain amounts of offset have been added to exactly separate them for good presentation."  It is not clear how much offset is added to each waveform.  Therefore, the denseness could very well be because of different offsets added to different waveforms.  In your algorithm you are using the sum of the coefficients.  Why not match the figure to the algorithm and plot the coefficient.  If you have a personal attachment to Figure 3 leave it there but add a figure with the coefficients.  In your mind you think Figure 3 is showing something; it is not.  At the very least add a figure which will show the grouping you are using in the algorithm.

3. Add a figure or another panel to Figure 4 to show the detail you want to emphasize about the ST segment.  You are perhaps young and have sharp eyes.  Your elderly readers like myself cannot see the details you are focusing on.  In the text explain the difference better.  Looking at the figure to my old eyes it seems that the R-wave signal has more extra components than the dct signal which I presume would be the ectopic beats.  Have you labeled the figure properly?  Something is wrong here.

Don't fight with the reviewer.  Use the review to improve the paper.

Author Response

Response to Reviewer 2 Comments

Point 1: The use of the word "modulus" is misleading in this context.  Use the phrase "absolute value" in Figure 2.  It will make your method more understandable.

Response 1: Thanks for reviewer’s advice, we had replaced all the misused word with the proper word.

Point 2: In Figure 3 you say that "These cosine waves could have been overlapped with each other, but certain amounts of offset have been added to exactly separate them for good presentation."  It is not clear how much offset is added to each waveform.  Therefore, the denseness could very well be because of different offsets added to different waveforms.  In your algorithm you are using the sum of the coefficients.  Why not match the figure to the algorithm and plot the coefficient.  If you have a personal attachment to Figure 3 leave it there but add a figure with the coefficients.  In your mind you think Figure 3 is showing something; it is not.  At the very least add a figure which will show the grouping you are using in the algorithm.

Response 2: Thanks for the advice, we added the DCT coefficient index based graph in Figure 3c. Figure 3c showed the basis about how to determine the optimal cut-point with calculating the gmin group which had the minimum value of amplitudes. Also, the word ‘dense’ was not proper in this researchs since we used the minimal summation value of coefficients, the description was refined to emphasize the cacluation by equation (6).

Point 3:. Add a figure or another panel to Figure 4 to show the detail you want to emphasize about the ST segment.  You are perhaps young and have sharp eyes.  Your elderly readers like myself cannot see the details you are focusing on.  In the text explain the difference better.  Looking at the figure to my old eyes it seems that the R-wave signal has more extra components than the dct signal which I presume would be the ectopic beats.  Have you labeled the figure properly?  Something is wrong here.

Response 3: We added Figure 5 for detailed comparison in Figure 4 and we also added two new references, ref. 31 and 32, to show why we only emphasize on ST-segment part because ST-segment elevation is one of the important features in ECG record with premature ventricular contractions, which was the one case we used from MIT-BIH record.